# Chemical linkers switch triglycerol detergents from bacterial protein purification to mild antibiotic amplification

Abhishek Kumar Singh [1,3] ✉, Marc Seewald [2,3], Boris Schade [1], Christian Zoister [1], Rainer Haag[1] & Leonhard Hagen Urner [2] ✉

Non-ionic detergents enable the investigation of cell membranes, including biomolecule purification and drug delivery. The question of whether non-ionic detergents associated with satisfying protein yields following extraction and affinity purification of proteins from lysed *E. coli* membranes can amplify antibiotics on whole-cell *E. coli* remains to be addressed. We unlock the modular chemistry of linear triglycerol detergents to reveal that more polar, non-ionic detergents that form globular micelles work better in amplifying antimicrobial activities of antibiotics than in purifying the membrane proteins mechanosensitive channel and aquaporin Z. Less polar detergents that form worm-like micelles indicate poor performances in both applications. With chromatography we demonstrate how fine-tuning the polarity of chemical linkers between detergent headgroups and tails can switch the utility of detergents from protein purification to antibiotic amplification. We anticipate our findings to be a starting point for structure-property studies to better understand detergent designs in supramolecular chemistry and membrane research.

Saccharide detergents are gold standards for the extraction of membrane components into globular micelles and enable the affinity purification and structural analysis of membrane proteins by X-ray crystallography or cryo-electron microscopy[1]. Oligoglycerol detergents (OGDs) are gold standards for the purification and native mass spectrometry analysis of membrane protein complexes[2–4], which includes the investigation of oligomeric states[5–7], top-down characterization through infrared multiphoton dissociation[8], drug binding[9], and structural role of lipid binding[5,7,10]. Seven triglycerol regioisomers are known (Fig. 1). Only two regioisomers have been translated into OGDs for membrane protein purification, i.e., triglycerol detergent regioisomers **a** and **b** (Fig. 1A)[11]. To expand possibilities for structure-property studies, herein, we establish the modular synthesis of triglycerol detergent regioisomer **c** (Fig. 1A, B).

The search for suitable detergents for protein purification has been explorative for decades[5,7]. Best detergents are identified empirically, leading to failing preparations and raising costs[7]. The development of detergents for protein purification is now moving towards a more rapid and intelligent design. For example, combinatorial synthesis[12], two-dimensional expansion synthesis[13], and Ugi-mediated detergent assembly[14] can improve the efficiency of synthesis and accessibility of chemical spaces through diversity-oriented screenings. Complementary, metric-assisted design approaches have been established[7,15]. Rather than synthesizing and testing detergents

randomly, the detergent properties most relevant to protein purification, like polarity, are tracked with metrices, like hydrophilic-lipophilic balance (HLB)[7,16]. Given a valid correlation in HLB and protein purification outcomes exists, the chemical space of the detergentome to be considered for synthesizing and testing can be narrowed down to a chemical space that likely contains suitable detergents for protein purification[7]. HLB values are calculated based on the structure of the head and tail[17,18], which are held together by a chemical linker (Fig. 1). The question if chemical linkers count as an extension of the polar head or non-polar tail remains to be clarified. We expect answering this question will improve accuracy of metric-assisted detergent design approaches and expand our capabilities in tuning detergent polarity for applications[12].

Furthermore, regardless of the strategy to be employed for the search of detergents, protein purification outcomes are compared with molecular detergent structures to establish detergent-class specific design guidelines[5]. The role of the detergent linker in aggregate morphology and protein purification outcomes has not yet been fully addressed. We expect that improving our knowledge on the link between detergent structure, supramolecular properties, and protein purification will reduce failure rates in detergent screenings and costs.

Structure-based drug discovery on membrane proteins supports the identification of drugs. Detergents can also amplify the effect of drugs on

[1]Institute of Chemistry and Biochemistry, Freie Universität Berlin, Berlin, Germany. [2]TU Dortmund University, Dortmund, Germany. [3]These authors contributed equally: Abhishek Kumar Singh, Marc Seewald. ✉e-mail: abhikmc@zedat.fu-berlin.de; leonhard.urner@tu-dortmund.de

**Fig. 1 | Establishing linear triglycerol detergents for structure-property studies. A** Schematic overview of all triglycerol regioisomers (left), triglycerol detergent regioisomers **a** and **b**, and triglycerol detergent regioisomer **c** whose synthesis and properties are subject of this work (right). **B** Overview of the research focus on triglycerol detergents in this work.

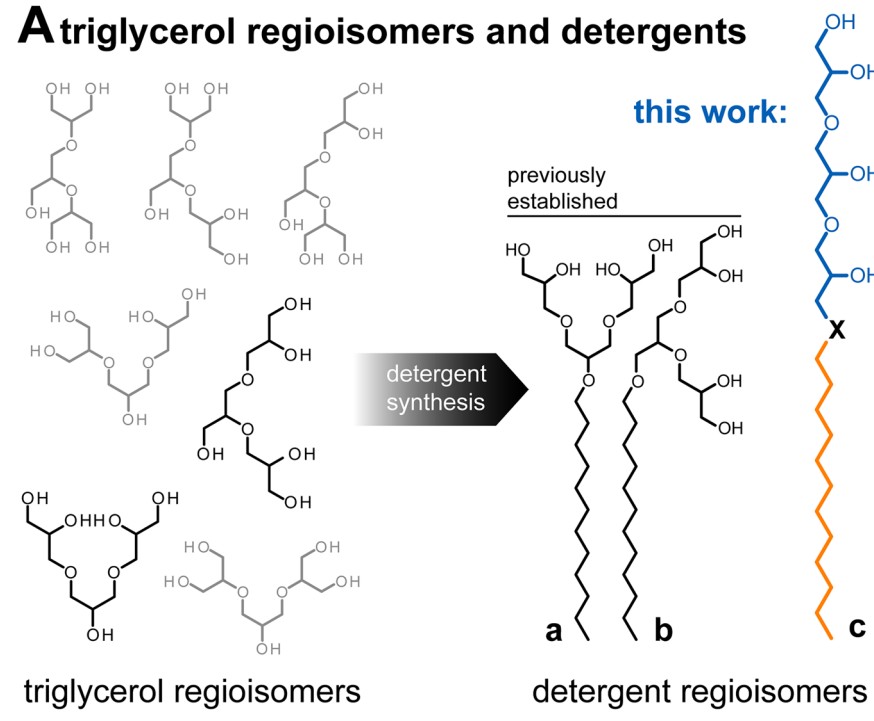

## A triglycerol regioisomers and detergents

triglycerol regioisomers

detergent regioisomers

**X** = thioether, ether, triazole, amide

## B research overview

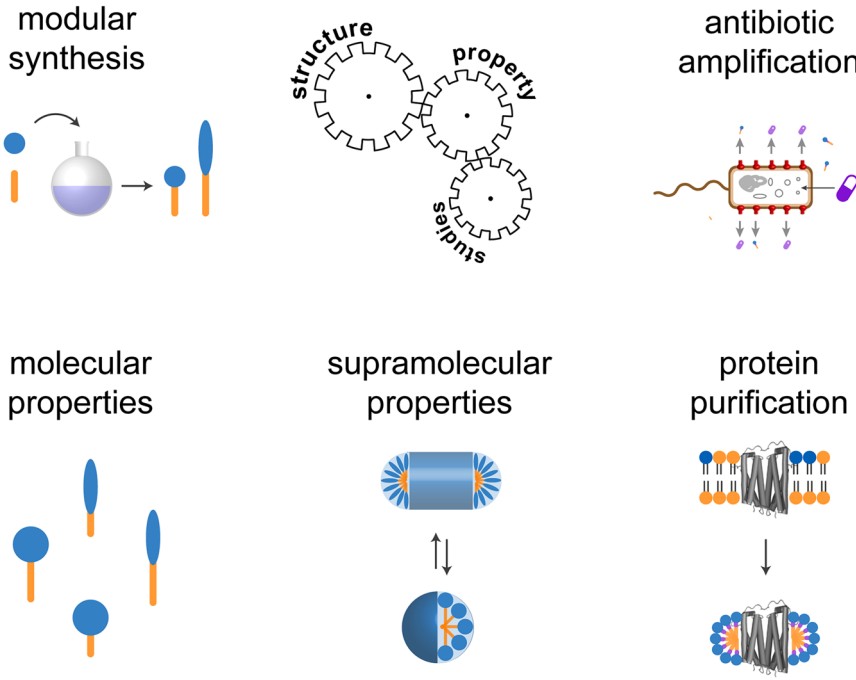

modular synthesis

antibiotic amplification

molecular properties

supramolecular properties

protein purification

cells directly by diminishing membrane integrity[19]. In the purification of inner or outer membrane proteins from *E. coli*, the solubilization of lysed membrane fragments with detergents is a key step to the extraction of high protein quantities into globular micelles[12,20,21]. Whether detergent properties associated with satisfying protein yields following extraction and affinity purification from lysed *E. coli* membranes translate into potent antibiotic amplification on whole-cell *E. coli* remains to be addressed. The correlation between the utility of non-ionic detergents for protein purification and drug

amplification is interesting for the development of detergent-containing drug formulations.

Here, we establish the modular chemistry of linear and dendritic triglycerol detergents to investigate how changes in detergent properties determine the correlation between supramolecular chemistry, extraction, and affinity purification of inner membrane proteins from lysed *E. coli* membranes and amplification of antimicrobial activity of antibiotics on whole-cell *E. coli*. Our results show that the assembly of non-ionic

detergents into more elongated, worm-like micelles indicates poor protein purification performance. The utility of non-ionic detergent micelles for protein purification correlates inversely with the utility for amplifying antimicrobial activities of antibiotics. Since aggregate morphologies are closely linked to detergent polarities, we develop a HPLC-based method to delineate how chemical linkers contribute to polar heads and nonpolar tails. Seen from a broader perspective, our results enable a better tuning of micellar detergent polarities and supramolecular properties for protein purification and underline the need for harsher detergents to overcome Gram-negative membranes, which represent an innovation hurdle in antibiotic research[22].

## Results

### Design of OGD Library

The common polar building blocks used for non-ionic detergent head groups are saccharides, polyethylene oxides, oligoglycerols, amino acids, and zwitter-ionic group, like fos-choline or amine oxides[1,3,20,21]. Detergents based on glycerol and related oligomers are distinguished by scalable water-solubility, scalable geometry, biocompatibility, and obtainability as a by-product of the vegetable oil industry[23–25]. Seven triglycerol regioisomers are known, which differ in terms of connectivity between triglycerol units, polarity, and shape (Fig. 1A)[26,27]. Synthesis protocols for related detergents have been established for two triglycerol regioisomers **a** and **b** (Fig. 1A)[11]. Interestingly, subtle differences in connectivity between glycerol units in detergent regioisomers **a** and **b** cause measurable changes in conical shape and polarity, which can improve membrane protein purification and stabilities of microfluidics droplets[5,11,28]. To expand possibilities for structure-property studies, herein, we establish the synthesis of less conically shaped, linear triglycerol detergent regioisomer **c** (Fig. 1A).

Structure-property studies generally benefit from establishing modular synthesis strategies that can tune the size and functionality of molecules in high yields from readily available starting materials[29–32]. All structural elements of OGDs, i.e., head, linker, tail, are relevant for supramolecular properties, protein purification, and amplifying antimicrobial activities of antibiotics[12,18,33,34]. To systematically investigate the role of head, linker, and tail, we designed a detergent library that contained linear and dendritic triglycerol heads (**a**, **c**), linkers with different polarities (thioether, ether, triazole, amide), and different alkyl tail lengths (C8, C12) (Fig. 2). Once we finished our library design, we set out to establish the modular chemistry of linear triglycerol detergent regioisomer **c**.

### Modular synthesis of linear and dendritic triglycerol detergents

To establish the modular synthesis of linear triglycerol detergent regioisomer **c**, we proposed allyl glycidyl ether is a suitable starting material (Fig. 2). Allyl glycidyl ether is a bifunctional molecule whose epoxide and double bond can be independently addressed. To establish the synthesis of linear triglycerol detergents containing an ether linker, i.e., **LTG-E-Cn**, we opened the epoxide in allyl glycidyl ether with solketal under basic conditions to obtain compound **1** (Fig. 2, left panel). Subsequent oxidation of the double bond and ring-opening reactions with deprotonated aliphatic alcohols led to the acetal-protected precursors for **LTG-E-Cn** (Fig. 2, left panel).

To synthesize linear triglycerol detergents containing a thiol linker, i.e., **LTG-S-Cn**, the epoxide in **2** was opened with thioacetic acid to obtain compound **3**. Subsequent hydrolysis of the thioester under acidic conditions, followed by alkylation through a thiol-ene click reaction led to the obtainment of **LTG-S-Cn** (Fig. 2, left panel).

To also enable the implementation of triazole and amide linkers, such as in the cases of **LTG-T-Cn** and **LTG-A-Cn**, the epoxide in compound **2** was opened with sodium azide to obtain compound **4** (Fig. 2, left panel). Subsequent copper-catalysed click reaction with alkyne-containing alkyl chains led to acetal-protected precursors for **LTG-T-Cn** (Fig. 2, left panel). Reducing the azide in compound **4** to an amine led to compound **5**, which we coupled with aliphatic carboxylic acids to obtain acetal-protected precursors for **LTG-A-Cn** (Fig. 2, left panel). Removal of the acetal-protecting

groups under acidic conditions led to the desired linear triglycerol detergents with C8 and C12 non-polar tails containing ether, amide, and triazole linkers, respectively (Fig. 2, left panel).

Having established the modular synthesis of OGD regioisomer **c**, we realized that the bifunctional character of allyl glycidyl ether enables the synthesis of OGD regioisomer **a** (Fig. 2). To explore an alternative route towards OGD regioisomer **a**, we opened the epoxide in allyl glycidyl ether with allyl alcohol under basic conditions to obtain compound **6** (Fig. 2, right panel). Subsequent oxidation with mCPBA, hydrolysis under acidic conditions, and treatment with 2,2-dimethoxy propane under acidic conditions led to acetal-protected triglycerol **8** (Fig. 2, right panel). Acetal-protected triglycerol **8** is a well-known starting material for the synthesis of dendritic triglycerol detergent regioisomer **a** containing ether, thiol, amide, or triazole linkers (Fig. 2). To finalize OGD regioisomers **a** proposed in this work, we utilized established procedures, like mesylation[35], azidation[35], reductive amination[36], etherification[11], thiol-ene click reaction[7], amide coupling[36], copper-catalysed azide-alkyne click reaction[35], and acetal deprotection (Fig. 2, right panel)[11]. The synthesis of OGD regioisomers **a** and **c** unlocks possibilities for structure-property studies.

### Protein purification and supramolecular chemistry

Membrane protein expression and purification is a tedious and resource-consuming process[37]. Reducing failure rates in the selection of potent detergent candidates for screenings is highly desired, because it ultimately reduces project costs. To investigate whether aggregate morphologies can be used to predict protein purification outcomes, we compared critical micelle concentration (cmc) values, aggregate morphologies obtained from our detergents above cmc (Supplementary Table 1) (Supplementary Fig. 1), and relative protein yields obtained upon extraction and affinity purification of the inner membrane protein mechanosensitive channel (MscL-GFP) (Fig. 3A–C). As reference, we included linear **DTG-E-C11**, a detergent that is also known as [G1] OGD[7–9] that delivers good protein quantities under the employed conditions.

We obtained two aggregate morphologies among our linear and dendritic triglycerol detergents, i.e., globular micelles and worm-like micelles (Fig. 3B). Regardless of the triglycerol head group, i.e., linear or dendritic, worm-like-micelle-forming detergents led consistently to lower relative protein quantities, suggesting that worm-like aggregates may be formed by detergents that deliver generally low relative protein quantities. However, a limitation of our approach is that the sample size of worm-like-micelle-forming detergents tested here is smaller ($n = 2$) compared to the micelle-forming detergents ($n = 7$).

To test the validity of our observation, we searched in the literature for relative protein yields obtained from other dendritic triglycerol detergents and re-investigated their aggregate morphologies[5,7]. Interestingly, similar reductions in relative protein quantities were obtained for another model protein, i.e., GFP-tagged aquaporin Z (AqpZ-GFP), when the C12-alkyl chain in dendritic triglycerol detergents was displaced by (a) a C14-alkyl chain and (b) a partially fluorinated chain[7]. Alternatively, a substitution of (c) a cholesterol tail in [G2] OGDs by a C12 double chain motif led to a similar observation (Supplementary Fig. 2)[5]. All three modification, i.e., (a)-(c), led to detergents that formed worm-like micelles[5,7,38]. Previously reported protein purification data on AqpZ-GFP[5,7,38,39] were done by following the same protocol as described here for MscL-GFP[39]. To exclude that our correlation is biased by the protein, herein, we repeated the purification of AqpZ-GFP under comparable conditions[39] with our detergents and obtained similar trends in relative protein quantities, which supports our conclusion that worm-like micelles correlate with low relative protein quantities (Supplementary Table 1) (Supplementary Fig. 2 and 3) (Fig. 3).

The question that remains to be addressed in this sub-chapter is whether aggregate morphologies can be used to predict protein purification outcomes. Since worm-like micelles correlate with low protein yields, we asked whether detergents that form globular micelles are automatically suitable for protein purification? The top 20 detergents used in protein purification indeed form globular micelles, which underlines that micelle-

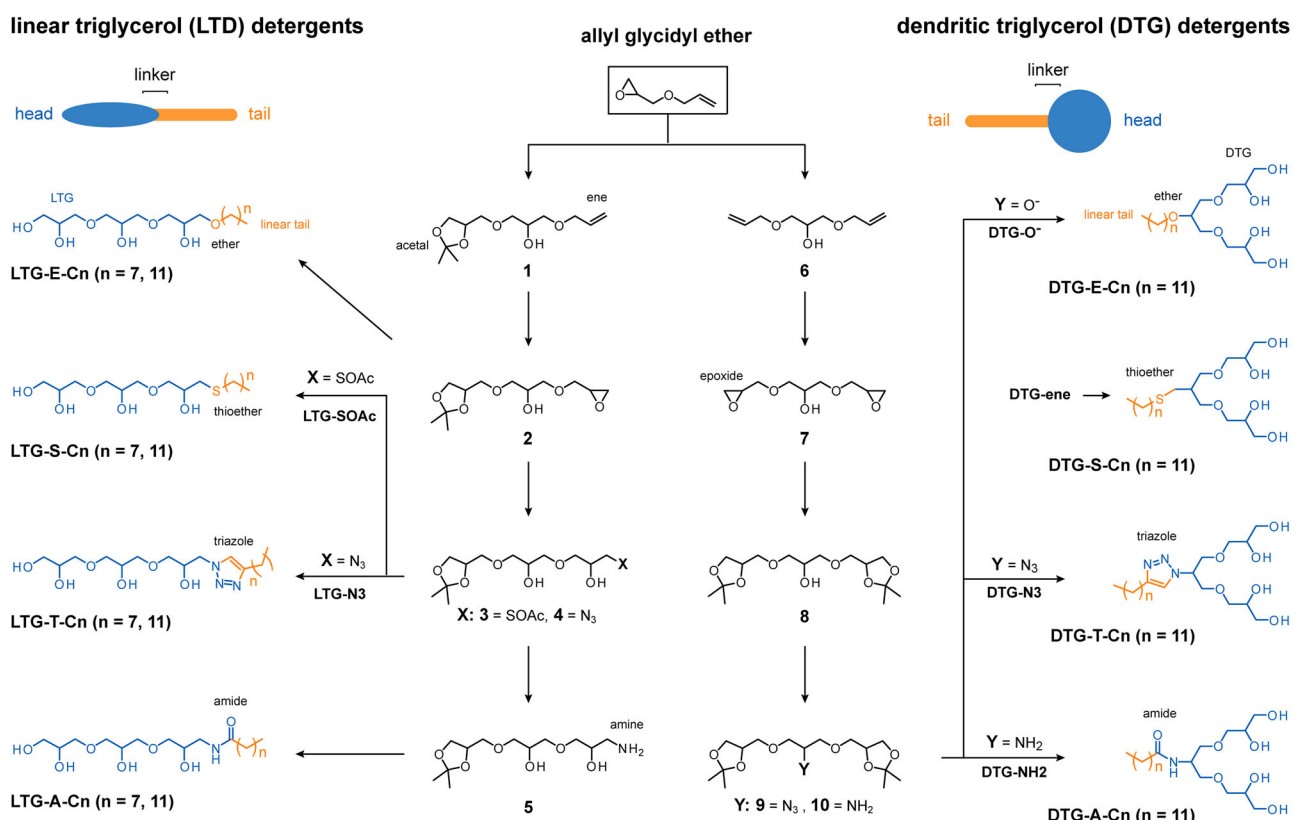

**Fig. 2 | Modular synthesis of linear and dendritic triglycerol detergents.** Overview of synthesis pathways starting from allyl glycidyl ether. Depending on the synthesis strategy, linear triglycerol detergents and dendritic triglycerol detergents are obtained, containing ether, thioether, triazole, or amide linkers between head and linear tail.

forming detergents are likely suitable for protein purification[7,40]. However, our data clarify also that detergents that assemble into globular micelles do not automatically enable protein purification. For example, we could practically obtain no relative protein yield with linear OGDs having shorter C8-alkyl chains (Fig. 3A). Our results confirm a consensus in the field. Detergents with shorter C8-alkyl chains have a reduced aggregation tendency, which increases cmc values and creates a harsh solution environment for proteins (Fig. 3B)[21,41]. The higher the detergent concentration in purification buffers, the more likely proteins denature[20,41]. Attempts to reduce the cmc of linear triglycerol detergents by installing a C12-alkyl chain led to worm-like micelle-forming detergents or practically water-insoluble detergents, which were not suitable for protein purification (Supplementary Table 1) (Fig. 3A). Among linear triglycerol detergents with C8-alkyl chains that assembled into globular micelles, only those with thioether and ether linkages could purify proteins (Fig. 3A).

Taken together, we conclude that due to their higher cmc values, linear triglycerol detergents are harsher detergents, compared to dendritic triglycerol detergents. Furthermore, aggregate morphologies can be used to estimate the success of detergents in protein purification. Worm-like-micelles indicate generally poor protein purification performance. The utility of detergents that form globular micelles is sensitive to detergent polarity and worth to be investigated further.

## Linker polarity and HLB values

Another emerging strategy to estimate the performance of globular, non-ionic detergent micelles in protein purification is the analysis of HLB values which are calculated from the molecular weights of polar headgroups and nonpolar tails. A central question raised in our study was whether our chemical linkers count as an extension of the polar head or nonpolar tail. To improve structural assignments for HLB analysis, we assessed the impact of linker chemistry on overall polarity of detergents by comparing isocratic elution profiles of linear triglycerol detergents with C8-alkyl chains by

reversed-phase chromatography (Fig. 4A). Retention times of detergents increased from amide < triazole < ether < thioether, which indicates that overall polarities decreased from amide to thioether (Fig. 4A).

To translate our results into HLB calculations, we assumed that the aliphatic ether linker is an extension of the non-polar tail (Fig. 4A). The ether linker is chemically similar to diethyl ether, which is a nonpolar and practically water-insoluble molecule[42]. Our HLPC data suggest that linker polarity is further reduced if oxygen is replaced with sulphur (Fig. 4A). The triazole linker exhibits amphiphilic properties and is more polar compared to the ether linker as reflected by a reduced retention time (Fig. 4A). Nitrogen atoms in triazole can be solvated and act as extension to the polar group, while the carbon double bond acts likely as extension to the non-polar tail[43]. In line with literature data[44], the amide linker is more polar than triazole and acts likely as an extension to the polar head group (Fig. 4A).

Having refined the contribution of the linkers to polar head groups and non-polar tails, we estimated differences in overall polarity by calculating HLB values (Fig. 4B)[12]. Based on our structural assignment, the HLB values of detergents increased in the direction from thioether < ether < triazole < amide, as exemplified for linear triglycerol detergents with non-polar C8 chains (Fig. 4B). Similar results were obtained with other linear and dendritic triglycerol detergents (Supplementary Table 1). This trend agrees with polarity differences obtained from HPLC measurements (Fig. 4A). As control, we calculated HLB values based on the assumption that all our linkers, per default, count as an extension of the non-polar tail. In this case, HLB values did not increase from thioether to amide (Fig. 4B). The fact that we could not recapitulate this control assumption in our HPLC measurements underlines that our refinement improved the connection between HLB model, detergent structure, and overall polarity (Fig. 4A, B).

## Antibiotic amplification on *E. coli*

The last question to be addressed here was if detergents that work well for protein purification also amplify drugs by diminishing membrane integrity.

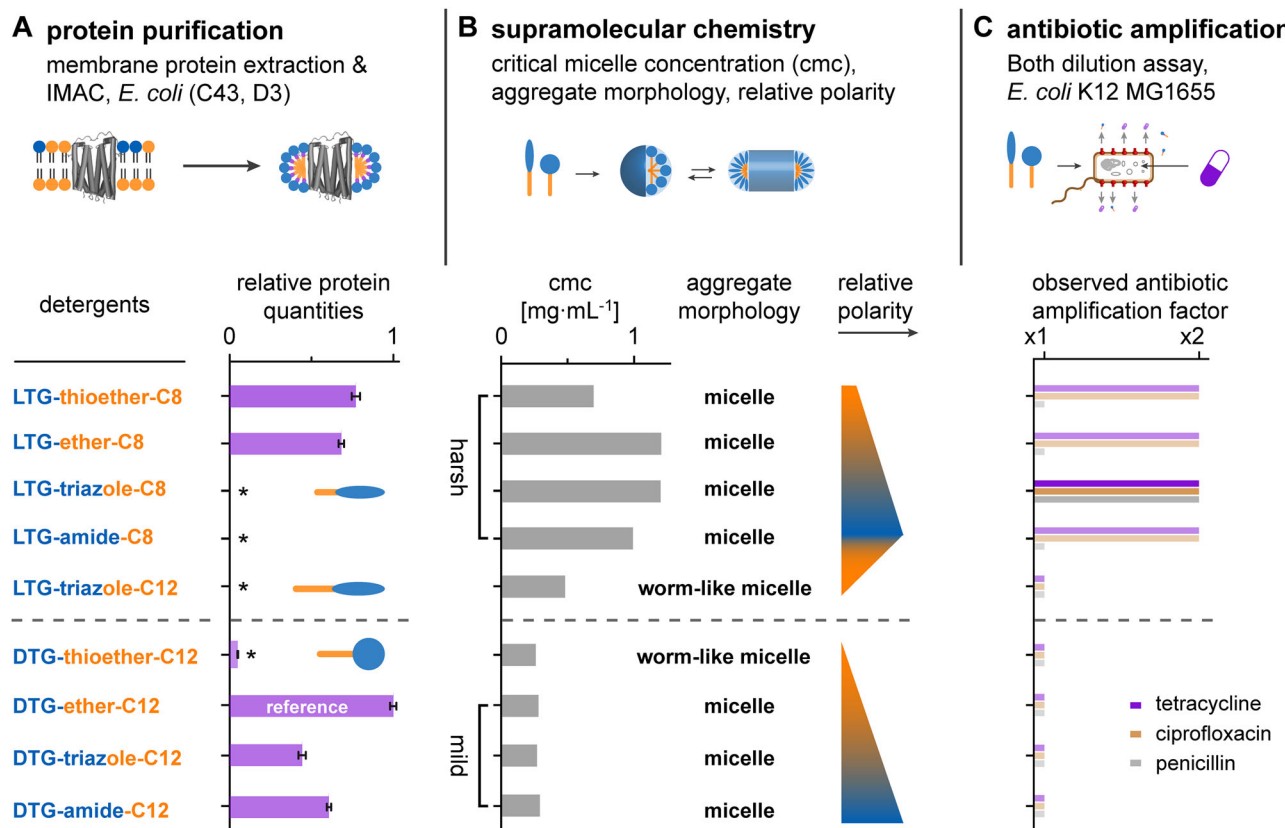

**Fig. 3 | Supramolecular chemistry affects protein purification and drug amplification. A** Bar chart (purple bars) showing relative protein quantities (±SE) obtained upon extraction and IMAC of bacterial MscL-GFP from lysed *E. coli* membranes with different detergents from two independent repeats (*n* = 2). Detergents with higher or lower cmc values are classified as harsh or mild.

**B** Overview of the detergents' cmc values (gray bars), aggregate morphologies and relative polarities (blue-orange bars). **C** Bar charts showing observed amplification factors for three antibiotics in the presence of 15x cmc detergent on whole-cell *E. coli* (1x = no amplification, 2x = MIC was halved). Source data are provided as Supplementary Data file.

To address this question, we investigated if our detergents could reduce the minimal concentration of antibiotics required to inhibit bacterial growth (MIC), including tetracycline, penicillin, and ciprofloxacin, by using the Broth microdilution assay (Supplementary Fig. 4).

We did protein purification experiments on lysed membrane fractions from *E. coli* (C43, DE3) and antibiotic amplification experiments on *E. coli* K12 MG1655. The situations faced during membrane solubilization with detergents in protein purification or antibiotic amplification experiments are vastly different. The cell wall of *E. coli* contains an outer membrane, peptidoglycan layer and an inner membrane (Fig. 5A). The outer leaflet of outer membranes consists of a robust lipopolysaccharide network, which is difficult to solubilize with non-ionic detergents and serves as a diffusion barrier with a molecular weight cut-off of approximately 600 Da (Fig. 5A)[22]. The inner membrane contains phospholipids and can be solubilized by non-ionic detergents, unless it is shielded by outer membranes (Fig. 5A)[45]. The robustness of *E. coli* cell walls is nicely reflected in our experimental observations. We found it generally difficult to obtain any amplification of antimicrobial activities of antibiotics by our non-ionic detergents, regardless of the detergent concentration (Fig. 3C). MIC values of antibiotics were halved by linear triglycerol detergents with linear C8-alkyl chains and only for the highest detergent concentrations tested under the employed conditions, i.e., 15x cmc (Fig. 3C). The only detergent that could halve the MIC of all three antibiotics was the linear triglycerol detergent with a triazole linker (Fig. 3C). This detergent is among the harshest detergents of our C8-alkyl chain derivatives. Its polar linker serves as extension of the polar head and non-polar tail, which is reflected in a high overall polarity, high cmc value, and poor protein purification performance (Fig. 3A–C). Our

findings suggest that non-ionic detergents that are harsher in protein purification work better in amplifying antimicrobial activities of antibiotics (Fig. 3A–C).

Detergents that are good for protein purification are generally good in solubilizing phospholipid membranes, which is linked to the fact that cells were lysed, and membranes fragmented (Fig. 5A)[46]. After lysis, the outer membrane no longer shields the inner membranes from detergents. Fragments of the inner membrane become readily accessible for detergent solubilization (Fig. 5A)[46]. Therefore, in protein purification, detergents that form globular micelles with lower cmc values can sufficiently release proteins from membranes (Fig. 3A, B). Harsher detergents are required to overcome intact outer membranes in *E. coli*. In line with our explanation, we observed stronger antibacterial properties from detergents that are generally harsher than non-ionic detergents, such as sodium dodecyl sulphate and dodecyltrimethylammonium bromide (Supplementary Fig. 5).

## Discussion
We established the modular chemistry of linear triglycerol detergents to tackle a set of timely questions regarding the utility of non-ionic triglycerol detergents for protein purification and amplifying antimicrobial activities of antibiotics. Regarding the first question that we could address with the modular chemistry of triglycerol detergents - Can aggregate morphologies be used to predict protein purification outcomes? - we can now say that the formation of worm-like micelles indicates poor protein purification performance (Fig. 3A, B). We expect that worm-like micelles correlate to low detergent polarities in such a way that detergent molecules preferably self-assemble with detergent molecules rather than with membrane components. Accordingly, we observed reduced water solubilities and consistently

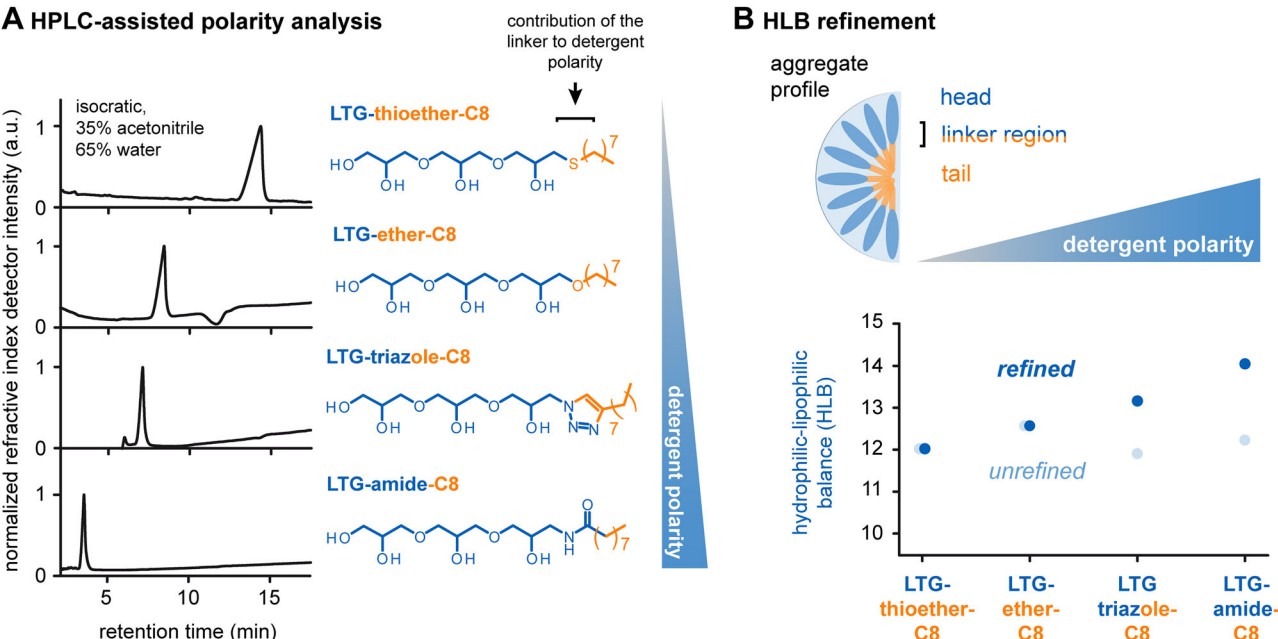

**Fig. 4 | HPLC-assisted linker assignment refinement. A** isocratic elution profiles of linear triglycerol detergents with C8-alkyl chains and different chemical linkers. **B** Comparison of the detergents' HLB values, before (transparent blue dots) and after

HPLC-assisted linker assignment refinement (blue dots). Dots shown for LTG-thioether-C8 and LTG-ether-C8, before and after refinement, were manually shifted to visualize the individual dots. Source data are provided as Supplementary Data file.

low protein yields for worm-like-micelle-forming detergents, beyond the compounds tested in this work (Supplementary Table 1) (Supplementary Fig. 2) (Fig. 3A, B). On the opposite, detergents that form globular micelles are no guarantee for successful protein purification. Careful tuning of polarities in globular micelles is required to balance improved water solubilities and protein compatibilities (Fig. 3A, B). Conveniently, detergents that assemble into worm-like micelles can be readily identified from globular micelles by their diffusion coefficients, for example, with dynamic light scattering, and removed from libraries prior to more expensive detergent screens in membrane protein purification (Supplementary Table 1).

Our conclusion that careful tuning of micellar polarities is required to optimize protein purification once more underlines the need for methodological improvements in tuning detergent polarities[7,34,47,48]. This led us to the second question that motivated this study: are chemical linkers (thioether, ether, triazole, amide) extensions of polar heads or non-polar tails? Our HPLC-assisted structural refinement suggests that thioether and ether linkers count as extensions of nonpolar tails. Nitrogen atoms in triazole extend polar headgroups while carbon double bonds extend nonpolar tails. In line with literature[44], the amide linker is more polar than triazole and counts as an extension of polar heads. Our HPLC-assisted structural refinement improved the link between linker polarity, detergent structure and HLB calculation, which will enable a better tuning of micelle polarities. More broadly, here-investigated chemical linkers are common building blocks in detergents and other amphiphiles[5,29,30,44], which will facilitate the future optimization of amphiphilic nanocarriers beyond this work.

The fact that non-ionic, detergents that form globular micelles frequently enable protein purification, because they efficiently solubilize phospholipid membranes, led us to the last question that motivated this work: do detergents with satisfying protein purification performance translate into potent drug amplifiers on cellular level? Our study highlights the ambivalence that detergents can be either (I) more suitable for amplifying antimicrobial activities of antibiotics or (II) membrane protein purification (Fig. 5B). From a supramolecular perspective, most detergents provide antimicrobial properties in their monomeric form, e.g., below cmc[49], because toxicity is linked to the insertion of monomeric detergents into membranes[50]. High cmc values lead to high monomer concentration around cmc whose uptake into membranes can facilitate structural

perturbations in lipid bilayers and the diffusion of antibiotics into bacteria. This is the opposite to what is required for protein purification where detergents are used above cmc to encapsulate membrane proteins into detergent aggregates that secure protein solubility and stability[21,41]. Detergents used in the extraction and affinity purification of membrane proteins ideally have low cmc values to minimize protein denaturation[3,21,41]. In line with this argumentation, our antibiotic amplification experiments on the model pathogen *E. coli* revealed that non-ionic detergents that are considered as "harsh" in protein purification, like detergents with shorter C8 alkyl chains and higher cmc values worked better in amplifying antimicrobial activities of antibiotics than detergents that are considered as "mild," like detergents with longer C12 alkyl chains and lower cmc values (Fig. 3A–C). Harsher detergents are more polar than milder detergents, which is reflected in reduced aggregation tendencies, higher cmc values, higher detergent monomer concentrations as well as higher overall detergent concentrations in assay buffers, which leads to more denaturing assay environments under the experimental conditions employed[21,41]. Established chemical design strategies that can maximize cmc values include a reduction in the overall size of detergent molecules, a reduction in the length of the alkyl tail relative to a polar head, or a substitution of non-ionic triglycerol headgroups for ionic headgroups[3]. Our data complement this knowledge by demonstrating that the utilities of equilibria between monomers and globular micelles formed by triglycerol detergents can be shifted from protein purification to mild amplification of antimicrobial activities of antibiotics by increasing cmc values through an increase in the polarity of the linker (Fig. 5B). In line with our suggestion that non-ionic, worm-like-micelle-forming detergents do not efficiently interact with membrane components, we did not observe amplifications in antimicrobial activities of antibiotics with such detergents (Fig. 5B).

Future studies need to clarify whether the incorporation of more polar linkers can also increase the cmc values and toxicities of other detergents classes. Further clarification is needed regarding the mode of action. Do detergents reduce the MIC of antibiotics more directly, for example, by promoting the diffusion of antibiotics into *E. coli*[51], or more indirectly, by destabilizing proton gradients across membranes and membrane protein machineries whose activities are coupled to proton gradients, such as observed before in the cases of certain amphiphilic efflux pump inhibitors[52].

**Fig. 5 | Outer membrane integrity determines solubilization. A** Schematic visualization of the cell wall of Gram-negative *E. coli*, including outer membrane (OM), peptidoglycan layer (PG), inner membrane (IM). Broken OMs, such as in the case of lysed *E. coli* (C43, D3) during protein purification, facilitate the solubilization of IMs by detergents. **B** Qualitative detergent assessment that compares utilities of triglycerol detergents in amplifying anti-microbial activities of antibiotics and protein purification in context with structural details, aggregate morphologies, and cmc values. Structural changes that go along with increased in linker polarities and cmc values shift the utility of triglycerol detergents between protein purification and mild antibiotic amplification.

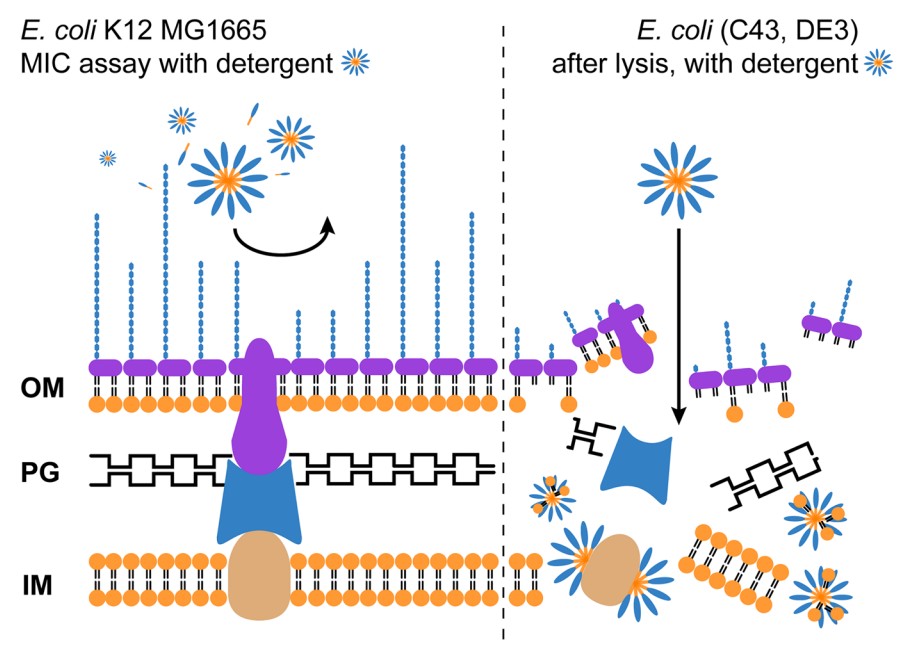

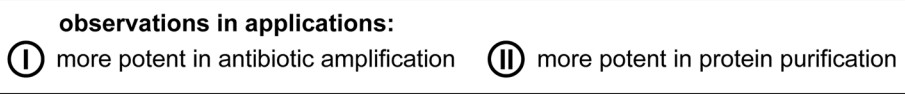

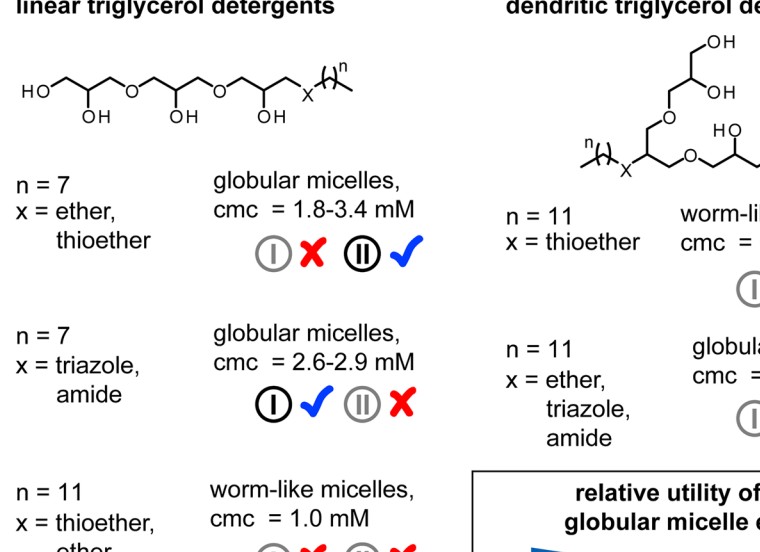

Future studies need to clarify if triglycerol detergents that efficiently solubilize lysed membrane fractions can synergize with reagents that solubilize outer membranes, such as benzalkonium chloride[53] or colistin[54]. Furthermore, since there is no such thing as a general cell wall structure, future studies need to contextualize the antibacterial properties of our detergents with different bacteria to better understand the impact of detergent design on selection pressure in microbial communities[19]. A better understanding on how the structures of detergents affect their selectivity to different cell wall architectures could deliver alternative directions for the development of cell-selective amphiphiles that serve the purposes of applications but do not harm the wider biosphere in their roles as novel entities[55].

In summary, we established the modular chemistry of linear and dendritic triglycerol detergents to deliver a complementary detergent class for membrane protein purification. We deliver a chromatographic method

for the specification of the detergent linkers (thioether, ether, triazole, amide) in functioning as an extension of the polar head and/or non-polar tail. These refined structural contributions enable a fine-tuning of linker polarities to switch the utility of harsher linear triglycerol detergents with shorter C8-alkyl chains between protein purification and mild amplification of antimicrobial activities of antibiotics. Our correlative study identified worm-like micelles as a general indicator for poor protein purification performance, which delivered an easy-to-implement approach based on dynamic light scattering to improve the detergent selection for membrane protein purification screens. Furthermore, our correlative study suggests that non-ionic detergents that are harsher in protein purification work better in amplifying antimicrobial activities of antibiotics due to higher polarities, cmc values and detergent monomer concentrations under assay conditions. Taking together, these learnings represent a significant step forward for the design of detergents in supramolecular chemistry, membrane research, drug delivery, and antibiotic formulations. The modular chemistry of triglycerol detergents and complementary assay technologies deliver starting points for structure-property studies that will be widely deployed beyond membrane research.

## Methods
### General information on detergent synthesis
All chemical reactions were performed in dried or distilled solvents for the synthesis. All commercially available compounds were purchased from Sigma-Aldrich (Germany), Acros Organics (Germany), Alfa Aesar (Germany), Fluka (Germany), Fischer Scientific (Germany), Merk (Germany), TCI (Germany), abcr Chemicals (Germany) and were used without any prior modifications. The solvents used for the reactions were dried and distilled before use. Other solvents, such as methanol (MeOH), dimethylformamide (DMF), dichloromethane (DCM), and tetrahydrofuran (THF), were used as supplied. Dry solvents were purchased in septum-sealed bottles from Sigma-Aldrich (Germany) and Acros Organics (Germany). To monitor the progress of the reaction, a Pre-coated TLC plate (Merck silica gel 60F254) was used with visualization of the spots on TLC using ceric and KMnO$_4$ stain solution. All synthesized compounds have been purified using column chromatography with silica gel (100–200 mesh) and NP automated column chromatography was done on a CombiFlash Rf (Teledyne ISCO) using prepacked silica columns (30 μm). NMR spectra were measured with an ECX 400 (400 MHz) and ECP500 (500 MHz) from JEOL and Avance 500 (500 MHz) or Avance 700 (700 MHz) NMR spectrometer from Bruker. Mass spectra were measured on a 6210 ESI-TOF and 6230 ESI-TOF from Agilent. The synthesis of all detergents discussed in this work in Fig. 1 is described in detail in the Supplementary Information, which is provided for download with the manuscript (Supplementary Schemes 1, 2 and 3). NMR data are provided in the Supplementary Information (Supplementary Figs. 7–18).

### Critical micelle concentration
Millipore water was used for the preparation of samples for their physico-chemical characterization and solubilization studies. The critical micelle concentration (cmc) was determined by means of an established solubilization assay including the model dye 'Nile red' and a fluorescence readout[56]. A stock solution of the dye in THF (1 mg·mL$^{-1}$) was prepared. Each 20 μL of this stock solution were added to 10 sample vials where the THF was allowed to evaporate to leave a thin film of the dye. Stock solutions of the amphiphiles in deionized water (5 mg·mL$^{-1}$) were stirred at least for 1 h before their serial dilutions to the final sample concentrations. The sample solutions were then transferred to the dye loaded vials and kept stirring overnight, before non-encapsulated dye was removed by filtration through 0.45 μm polytetrafluoroethylene (PTFE) filter. Fluorescence measurements were performed using a Cary Eclipse fluorescence spectrophotometer. To graphically determine the cmc, the fluorescence intensity at $\lambda = 635$ nm was plotted against the amphiphile concentration. Two linear regions were obtained, i.e., below and above the cmc. Both regions were fitted to linear functions and the intersection was taken as cmc (Supplementary Fig. 19).

### Diffusion coefficient analysis by dynamic light scattering
To determine diffusion coefficients of aggregates formed by detegents above cmc, dynamic light scattering (DLS) measurements were performed[57]. We used a Malvern Zetasizer Nano ZS analyser with temperature-controlled sample chamber and 4 mW He-Ne laser ($\lambda = 633$ nm) with back scattering detection (scattering angle $\theta = 173°$) and an avalanche photodiode as a detector. Disposable micro–BRAND UV-Cuvettes were used for the measurements. Aqueous detergent solutions (concentration = 2.5 mg·mL$^{-1}$) were prepared by constant stirring for 24 h at room temperature (~22.5 °C). All measurements were repeated three times with ten runs per measurement. Disposable *BRAND* UV-Cuvette's were used. Diffusion coefficients and, if applicable, hydrodynamic radii were extracted from DLS data as summarized in Supplementary Table 1.

### Cryogenic electron microscopy
To monitor aggregate morphologies, aqueous detergent solutions (concentration = 2.5 mg·mL$^{-1}$) were investigated by cryogenic electron microscopy. Perforated carbon film covered microscopical 200 mesh grids (R1/4 batch of Quantifoil, MicroTools GmbH, Jena, Germany) were cleaned with chloroform and hydrophilized by 60 s glow discharging at 10 μA in a EMSCOPE SC500 before 4 μl aliquots of the amphiphile solution were applied to the grids. The samples were vitrified by automatic blotting and plunge freezing with a FEI Vitrobot Mark IV (Thermo Fisher Scientific Inc., Waltham, Massachusetts, USA) using liquid ethane as cryogen and then transferred to the autoloader of a FEI TALOS ARCTICA electron microscope (Thermo Fisher Scientific Inc., Waltham, Massachusetts, USA). The microscope was equipped with a high-brightness field-emission gun (XFEG) that was operated at an acceleration voltage of 200 kV. Micrographs were acquired on a FEI Falcon 3 direct electron detector (Thermo Fisher Scientific Inc., Waltham, Massachusetts, USA) at a nominal magnification of 28,000, corresponding to a calibrated pixel size of 3.75 Å per pixel.

### Analytical HPLC measurements
To compare the overall polarities of detergents by means of their retention times under isocratic conditions[11], an AZURA® analytical HPLC system from Knauer was used, including an AZURA® pump P 6.1 L, AZURA® refractive index detector RID 2.1 L, an Autosampler AS 6.1 L, and an AZURA® column thermostat CT 2.1. As stationary phase a pre-packed RSC-Gel C18ec column was used (pore size: 100 Å, particle size: 5 μm, length 125 mm, diameter: 4 mm), purchased from Sauerbrey (Germany). The following experimental parameters were used to monitor retention times of detergents: temperature of the column oven: 35 °C; isocratic solvent mixture: acetonitrile 35% and water 65%; flowrate: 1 mL·min$^{-1}$; sample concentration: 6 mg·mL$^{-1}$; injection volume: 50 μL. Chromatogram data were exported as text files (intensity vs retention time), imported and plotted with OriginProV9.1, exported as Adobe Illustrator file, finalized by means of Adobe Illustrator CS2, exported and included into the manuscript as PNG file.

### Hydrophilic-lipophilic balance
To estimate changes in overall polarity, hydrophilic-lipophilic balance (HLB) values of detergents were calculated using the procedure established by Griffin[17]. Briefly, the HLB values of detergents were calculated using Eq. (1) in which individual parameters are defined as follows: molecular weight of the tail (MWtail) and molecular weight of the detergent (MW). HLB values are summarized in Supplementary Tables 1 and 2.

$$HLB = 20 \cdot \left(1 - \frac{MW_{tail}}{MW}\right) \qquad (1)$$

### Extraction and affinity purification of membrane proteins
Membranes containing overexpressed MscL-GFP and AqpZ-GFP were prepared from *E. coli* (C43, D3) (purchased from Cambridge Bioscience) as described in detail before[39]. All steps were done at 4 °C or on ice. To purify

proteins, membrane aliquots containing either overexpressed, His-tagged MscL-GFP or Aqpz-GFP (100 µL) were mixed with extraction buffer (700 µL of 20 mM Tris, 200 mM NaCl, pH = 8) and aqueous detergent solution (100 µL of a 10w% w/v solution in deionized water). The mixtures were agitated and incubated for 10 min. The supernatant was clarified by centrifugation (10,000 $g \times 10$ min) and purified by immobilized metal affinity chromatography (IMAC) using empty Biospin columns that were loaded with 800 µL Ni-NTA resin suspension (Qiagen). So-obtained IMAC columns [1 column volume (CV) = 400 µL] were washed with water (2 CVs), and IMAC wash buffer (2 CVs of 20 mM Tris, 200 mM NaCl, 20 mM imidazole, 2x cmc detergent of interest, pH = 8). The clarified supernatant was loaded on the IMAC column and the flow-through was discarded. The IMAC column was washed with IMAC wash buffer (2 CVs), IMAC pre-elute buffer (5 CVs of 20 mM Tris, 200 mM NaCl, 40 mM imidazole, 2x cmc detergent of interest, pH = 8) and the flow-through was discarded. Proteins were eluted with IMAC elute buffer (1 CV of 20 mM Tris, 200 mM NaCl, 200 mM imidazole, 2x cmc detergent of interest, pH = 8). The protein solutions were collected in plastic cuvettes and the absorption at 485 nm was determined by UV/Vis spectroscopy ($A_{485}$). The $A_{485}$ values were normalized to the values obtained from DTG-ether-C12 and the averages from two independent repeats ($n = 2$) were plotted with standard error of the mean (±SE) against detergent abbreviations (Supplementary Fig. 3) (Fig. 3A). Relative protein purities were confirmed by SDS PAGE analysis (Supplementary Fig. 6).

### Broth microdilution assay

For MIC tests, we used *E. coli* K12 MG1655 [M. S. Guyer strain, purchased from the DSMZ (Germany), DSM No.: 18039]. To determine MIC values of detergents, antibiotics and/or detergent-antibiotic combinations, we used the following procedure[58,59]: for preparation of sterile agar plates, a mixture of agar-agar (1.7 g), Müller Hinton Broth medium (MH medium) (2.1 g), and deionized water (100 mL) was autoclaved, poured into sterile petri dishes, and allowed to solidify at room temperature. For the preparation of individual bacterial colonies, a mixture of MH medium (4.2 g) and deionized water (200 mL) was autoclaved. A glycerol stock solution of *E. coli* MG1655 (750 µL) was dissolved in 5 mL sterile MH medium and incubated for 3.5 h at 37 °C. Different dilutions were prepared from the incubated suspension (1:500 & 1:1000 & 1:10,000, v:v). From each dilution, a 50 µL aliquot was pipetted onto a sterile agar plate. The suspensions were spread with a sterile L-shaped cell spreader and the agar plates were incubated at 37 °C for 16 h to obtain individual colonies.

To determine MIC values of detergents and antibiotics, detergent and antibiotic stock solutions were prepared in sterile MH medium. Furthermore, a single colony was transferred into 5 mL sterile MH medium. The mixture was shaken at 37 °C with 180 rpm. After 3.5 h the optical density at 600 nm of the suspension was determined using disposable cuvettes and a NanoPhotometer (IMPLEN). Bacterial suspension was diluted with sterile MH medium to a concentration of $1 \times 10^8$ cfu·mL$^{-1}$ (OD$_{600}$ = 0.06). A 1:100 v:v dilution was created in sterile MH medium to prepare the inoculum for the MIC test. An automatic 8-channel pipette from Eppendorf (Eppendorf Research plus 1200 µL) was used to pipette sterile MH medium (100 µL) into the wells 1–10 and 12 (growth control) of a sterile 96-well plate. For a sterile control, sterile MH medium (200 µL) was pipetted into the 11th well. Next, detergent stock solution (100 µL of 1x cmc DTAB, or 141x cmc SDS, or 46.25x cmc LTG-ether-C8, or 198.21x cmc DTG-triazole-C12) or antibiotic stock solution (100 µL of 1.56 µg/mL TET, or 250 µg/mL PCN, or 31.25 ng/mL CFX) was added to the first well and the solution within the well was mixed by pipetting seven times up and down. Then 100 µL of the solution in the first well was transferred to the second well and mixed again as described for the first well before. This procedure was repeated until reaching the 10th well. The remaining 100 µL taken from the 10th well were discarded. No detergent was added to the 11th and 12th well. Subsequently, *E. coli* inoculum (100 µL) was added to the wells 1–10 and 12. The 96-well plate was incubated at 37 °C overnight. The lowest detergent or antibiotic concentration at which no visible bacterial growth occurred was taken as MIC.

To determine antibiotic amplification factors of detergents, antibiotic stock solutions (1.56 µg/mL TET, or 250 µg/mL PCN, or 31.25 ng/mL CFX) were prepared in detergent-containing sterile MH medium (30x cmc of detergent of interest). Bacterial suspension was prepared as described before. An automatic 8-channel pipette from Eppendorf (Eppendorf Research plus 1200 µL) was used to pipette detergent-containing, sterile MH medium (100 µL, 30x cmc of detergent of interest) into the wells 1–10 of a sterile 96-well plate. For a sterile control and growth control, sterile MH medium (200 µL & 100 µL) was pipetted into the 11th and into the 12th well. Next, antibiotic stock solution that also contained detergent (100 µL of 1.56 µg/mL TET, or 250 µg/mL PCN, or 31.25 ng/mL CFX with 30x cmc detergent of interest, respectively) was added to the first well and the solution within the well was mixed by pipetting seven times up and down. Then 100 µL of the solution in the first well was transferred to the second well and mixed again as described for the first well before. This procedure was repeated until the 10th well. The remaining 100 µL taken from the 10th well were discarded. No detergent-antibiotic mixture was added to the 11th and 12th well. Subsequently, *E. coli* inoculum (100 µL) was added to the wells 1–10 and 12. The 96-well plate was incubated at 37 °C overnight. The lowest antibiotic concentration at which no visible bacterial growth occurred was taken as MIC. To determine the antibiotic amplification factor of detergents, the MIC value of the antibiotic alone was divided by the MIC value of the detergent-antibiotic combination.

### Reporting summary

Further information on research design is available in the Nature Portfolio Reporting Summary linked to this article.

### Data availability

Datasets generated during and/or analysed during the current study are available from the corresponding authors on reasonable request.

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

## Acknowledgements
We thank the core-facility BioSupraMol of the Freie Universität Berlin for the analytical support. The Ministry of Culture and Science of the German State of North Rhine-Westphalia (NRW return program), North Rhine-Westphalian's Academy of Sciences, Humanities, and the Arts of the German State of North Rhine-Westphalia (Junges Kolleg), and Fonds der Chemischen Industrie (material cost allowance) are gratefully acknowledged for financial support.

## Author contributions
L.H. Urner and A.K. Singh conceptualized the work. A.K. Singh, M. Seewald, B. Schade, C. Zoister, and R. Haag synthesized and/or characterized the detergents. L.H. Urner and A.K. Singh purified proteins. L.H. Urner and M. Seewald performed antibacterial tests. L.H. Urner wrote the manuscript with input from all authors.

## Funding

## Competing interests
The authors declare no competing interests.
