## [Peer review file · Communications Chemistry]

Chemical linkers switch triglycerol detergents from bacterial protein purification to mild antibiotic amplification

Corresponding Author: Dr Leonhard Hagen Urner

Version 0:

Reviewer comments:

Reviewer #1

(Remarks to the Author)

The authors investigate how chemical linkers in newly designed detergent molecules influence their effectiveness in protein purification and their ability to enhance the efficacy of antibiotics. They synthesized a library of triglycerol detergents with various head groups, linkers, and tails to systematically explore their properties. The study emphasizes that careful adjustment of micellar polarity is crucial for optimizing protein purification. To achieve this, the authors developed an HPLC-based method to quantify detergent polarity and demonstrated an inverse correlation between a detergent's suitability for protein purification and its ability to amplify antibiotic effects.

This reviewer notes two significant strengths in this manuscript. First, it presents a modular chemistry approach for synthesizing linear and dendritic triglycerol detergents to systematically investigate structure–property relationships. Second, the discovery of an inverse relationship between suitability for protein purification and “antibiotic amplification” is a novel finding. This work contributes to a fundamental understanding of detergent chemistry and provides valuable tools for membrane-protein research and drug discovery.

However, the manuscript has two major areas for improvement, namely, the presentation of original data and the Discussion section. While it includes chemical structures, reaction schemes, and bar charts, there is minimal original data beyond the HPLC elution profiles in Figure 4A. To provide insight into data quality and the robustness of the conclusions, the authors should include at least a few representative examples of primary data for each type of experiment, even if these are considered “standard.” Additionally, the Discussion section largely repeats key results without adequately engaging with previous studies and the substantial body of literature on detergent design and applications in protein science and as antimicrobial agents.

Minor Issues:

- Abstract, second line: Remove the hyphen after “molecular.”
- The terms “drug amplification” and “antibiotic amplification” and phrases like “amplify drugs” are used imprecisely. The authors should define and clarify this concept, particularly by referring to it as “amplification of drug efficacy” or more specifically, “amplification of antimicrobial activity,” to improve clarity.
- Introduction, first paragraph: Define “IRMPD-based.”
- Introduction, second paragraph: The phrase “metric-assisted design approaches are emerging” should acknowledge that metrics like the hydrophobic-lipophilic balance (HLB) have been established and in use for years, so rational design is not entirely new to detergent development.
- Figure 1A: The letter “c” denoting the figure panel is much larger than “a” and “b”.
- Figure 1B: The cogwheels do not interact, which likely does not convey the intended meaning.
- Throughout the manuscript, it would be clearer to differentiate between “globular micelles” and “worm-like micelles” (or “spheroidal micelles” and “cylindrical micelles”) rather than simply between “micelles” and “worm-like micelles,” as worm-like micelles are still a type of micelle.
- Page 14, second paragraph: The statement “Detergents that are good for protein purification are generally good at

solubilizing phospholipid membranes" is an oversimplification. For example, SDS effectively solubilizes lipid membranes but would denature proteins, making it unsuitable for protein purification. Conversely, DDM is a mild detergent commonly used for protein extraction and purification but is a relatively poor solubilizer of lipid membranes.

Reviewer #2

(Remarks to the Author)

General Comments:

In the manuscript submitted by Singh et al., it was developed a modular chemistry of linear and dendritic triglycerol detergents to deliver a new detergent class for membrane protein purification, and drug amplification. The subject of the manuscript is interesting for a broad variety of readers, with potential significance in the field of membrane protein research and drug development. The manuscript is well written and organized. However, the originality of this manuscript should be better highlighted.

Specific Comments

- The abstract should be improved in terms of obtained results.
- Specific terms, such as "worm-like micelles," are not always explained in depth for non-specialists in this field. A brief description would be helpful.
- It would be interesting to include a table comparing the detergents' properties (e.g., HLB values, protein yield, and MIC reductions) of this work with those from published works.
- The discussion section can be more concise in terms of polarity effects on detergent performance, since they are repeated in some parts without adding new insights.
- Expand the potential reasons why harsh detergents perform better for antibiotic amplification but not for protein purification.
- Lack of critical comparative analysis. While this manuscript cites previous studies, it lacks direct comparative data to emphasize novelty and contextualize findings.
- Discuss potential limitations or next steps, such as scaling your methodology or testing against other Gram-negative pathogens
- In the conclusion section it would be interesting to summarize how the findings support future applications (e.g., drug delivery? or antibiotic formulations?).

Reviewer #3

(Remarks to the Author)

Manuscript: Chemical linkers switch triglycerol detergents from bacterial protein purification to mild antibiotic amplification
The English language has to be improved through the manuscript.

The authors discuss protein purification and mention membrane dissolution in the introduction section. However, they need to specify the type of protein they aim to explore—whether it is produced intracellularly by the microorganism or secreted into the culture medium. Additionally, different microorganisms have varying types of cell walls, which require distinct treatments and interactions with detergents for membrane disruption. I suggest revising the introduction section to clarify these points and provide more detailed explanations.

The size of the detergent tail may influence its interaction with the cell wall and the target molecule intended for purification. How does the authors' approach address or analyze this potential impact?

The authors determined the critical micelle concentration (CMC) of the detergents, and I suggest including these results in a table for better clarity. Additionally, are there alternative methods to determine this parameter besides fluorescence? If so, it would be useful to discuss them.

I recommend expanding the discussion section to highlight key factors that can guide the purification of a molecule using detergents. There are studies in the literature that report the extraction of molecules in micellar systems, yet the authors consistently use the term "purification" throughout the manuscript. Was this study designed to be applicable to other detergent families? What challenges might arise when using different compounds?

Furthermore, it would be important to clarify whether the detergents used in this study can remain associated with the molecule for its intended application, or if they must be removed. If removal is necessary, what procedures could be employed to achieve this effectively?

Version 1:

Reviewer comments:

Reviewer #1

(Remarks to the Author)

The authors have addressed all comments in a convincing manner, so I recommend publication of the revised manuscript.

Reviewer #2

(Remarks to the Author)

The authors have addressed the comments in the revised manuscript version.

I am satisfied with their response and believe that the revised manuscript can now be accepted.

Reviewer #3

(Remarks to the Author)

The authors made several changes in the manuscript according to reviewer suggestions. The topics of the manuscript are interesting for readers from different fields. My suggestion is for manuscript acceptance in the present format.

Reviewers' comments:

Reviewer #1 (Remarks to the Author):

1.) The authors investigate how chemical linkers in newly designed detergent molecules influence their effectiveness in protein purification and their ability to enhance the efficacy of antibiotics. They synthesized a library of triglycerol detergents with various head groups, linkers, and tails to systematically explore their properties. The study emphasizes that careful adjustment of micellar polarity is crucial for optimizing protein purification. To achieve this, the authors developed an HPLC-based method to quantify detergent polarity and demonstrated an inverse correlation between a detergent's suitability for protein purification and its ability to amplify antibiotic effects.

This reviewer notes two significant strengths in this manuscript. First, it presents a modular chemistry approach for synthesizing linear and dendritic triglycerol detergents to systematically investigate structure–property relationships. Second, the discovery of an inverse relationship between suitability for protein purification and “antibiotic amplification” is a novel finding. This work contributes to a fundamental understanding of detergent chemistry and provides valuable tools for membrane-protein research and drug discovery.

However, the manuscript has two major areas for improvement, namely, the presentation of original data and the Discussion section. While it includes chemical structures, reaction schemes, and bar charts, there is minimal original data beyond the HPLC elution profiles in Figure 4A. To provide insight into data quality and the robustness of the conclusions, the authors should include at least a few representative examples of primary data for each type of experiment, even if these are considered “standard.”

First of all, we would like to thank all referees for their comments which helped us to improve clarity, and the scientific value of our manuscript.

To provide insight into data quality and the robustness of the conclusions, we added the following primary data for each type of experiment:

- SDS PAGE gels showing characteristic protein bands for MscL-GFP and AqpZ-GFP to confirm protein IDs and comparable purity of protein preparations (see Supplementary Figure 2)
- HLB calculation table including detergent molecular weight (MW), refined/unrefined molecular weights of detergent tails (MW_{tail}), and refined/unrefined HLB values (see Supplementary Table 2)
- ¹H/¹³C NMR data of final detergents (see Supplementary Figures 7-18)
- representative fluorescence data for cmc determination (see Supplementary Figure 19)
- a description of the protein purification experiments was added to the Method section

2.) Additionally, the Discussion section largely repeats key results without adequately engaging with previous studies and the substantial body of literature on detergent design and applications in protein science and as antimicrobial agents.

We thank the referee for raising this point, which helped us to better contextualize our work with literature and gave us new insights, which we communicated in the revised manuscript as outlined below.

First, we expanded our discussion in the revised manuscript to communicate our knowledge gain related to polarity effects on detergent performance – for details, please see answer to reviewer comment No.14.

Second, we expanded our discussion in context with literature on the potential reasons why harsh detergents perform better for antibiotic amplification but not for protein purification – for details, please see answer to reviewer comment No.15.

Third, we expanded our discussion on potential limitations or next steps in context with the development of antimicrobial reagents – for details, please see answer to reviewer comments No. 17 and 18.

Finally, we specified our case in the introduction which improved the frame of our story line – for details, please see answer to reviewer comments. 19 and 23. #

3.) Minor Issues:

- Abstract, second line: Remove the hyphen after “molecular.”

We removed the hyphen.

4.) The terms “drug amplification” and “antibiotic amplification” and phrases like “amplify drugs” are used imprecisely. The authors should define and clarify this concept, particularly by referring to it as “amplification of drug efficacy” or more specifically, “amplification of antimicrobial activity,” to improve clarity.

We specified all relevant phrases in the revised manuscript accordingly, for example, by referring more specifically to it as “amplifying the antimicrobial activities of antibiotics.”

5.) Introduction, first paragraph: Define “IRMPD-based.”

We replaced the abbreviation IRMPD as follows: “infrared multiphoton dissociation,⁸”

6.) Introduction, second paragraph: The phrase “metric-assisted design approaches are emerging” should acknowledge that metrics like the hydrophobic-lipophilic balance (HLB) have been established and in use for years, so rational design is not entirely new to detergent development.

To address this point, we revised the sentence as follows: “Complementary, metric-assisted design approaches have been established.^{7, 15}”

7.) Figure 1A: The letter “c” denoting the figure panel is much larger than “a” and “b”.

The denotings of “a,” “b,” and “c” are now similar in size.

8.) Figure 1B: The cogwheels do not interact, which likely does not convey the intended meaning.

We put the cogwheels together to make them interact.

9.) Throughout the manuscript, it would be clearer to differentiate between “globular micelles” and “worm-like micelles” (or “spheroidal micelles” and “cylindrical micelles”) rather than simply between “micelles” and “worm-like micelles,” as worm-like micelles are still a type of micelle.

We now specified the terms globular micelles and elongated, worm-like micelles throughout the revised manuscript.

10.) Page 14, second paragraph: The statement “Detergents that are good for protein purification are generally good at solubilizing phospholipid membranes” is an oversimplification. For example, SDS effectively solubilizes lipid membranes but would denature proteins, making it unsuitable for protein purification. Conversely, DDM is a mild detergent commonly used for protein extraction and purification but is a relatively poor solubilizer of lipid membranes.

To avoid this oversimplification, we revised our statements in the abstract as follows: “Non-ionic detergents enable the investigation of cell membranes, including biomolecule purification and drug delivery. The question of whether non-ionic detergents associated with satisfying protein yields following extraction and affinity purification of proteins from lysed *E. coli* membranes can amplify antibiotic amplification on whole-cell *E. coli* remains to be addressed.”

Reviewer #2 (Remarks to the Author):

General Comments:

In the manuscript submitted by Singh et al., it was developed a modular chemistry of linear and dendritic triglycerol detergents to deliver a new detergent class for membrane protein purification, and drug amplification. The subject of the manuscript is interesting for a broad variety of readers, with potential significance in the field of membrane protein research and drug development. The manuscript is well written and organized. However, the originality of this manuscript should be better highlighted.

Specific Comments

11.) The abstract should be improved in terms of obtained results.

We revised the abstract in context with our updated scientific question and emphasis on obtained results as follows: “Non-ionic detergents enable the investigation of cell membranes, including biomolecule purification and drug delivery. The question of whether non-ionic detergents associated with satisfying protein yields following extraction and affinity purification of proteins from lysed *E. coli* membranes can amplify antibiotics on whole-cell *E. coli* remains to be addressed. We unlock the modular chemistry of linear triglycerol detergents to reveal that more polar, non-ionic detergents that form globular micelles work better in amplifying antimicrobial activities of antibiotics than in membrane protein purification. Less polar detergents that form elongated, worm-like micelles indicate generally poor performances in both applications. We use chromatography to demonstrate how fine-tuning the polarity of chemical linkers between detergent headgroups and tails can switch the detergent’s utility from protein purification to antibiotic amplification. We anticipate our findings to be a starting point for structure-property studies to better understand the design of detergents in supramolecular chemistry and membrane research.”

12.) Specific terms, such as "worm-like micelles," are not always explained in depth for non-specialists in this field. A brief description would be helpful.

We now specified the term as “elongated, worm-like micelles” throughout the revised manuscript.

13.) It would be interesting to include a table comparing the detergents' properties (e.g., HLB values, protein yield, and MIC reductions) of this work with those from published works.

We thank the referee for raising this point. We find it difficult to combine all data since detergents published previously did not include MIC tests. Furthermore, in terms of protein purification, we refer to two data sets (see Ref.1 and Ref.2 in Supporting Information) that were obtained using the same assay protocol but compared to different reference detergents, i.e., DTG-ether-C12 in Ref.1 or DDM from Ref.2. This makes a direct comparison of both data sets difficult. However, the relative comparison of protein yields within each data set supports our conclusion that detergents which assemble into worm-like micelles correlate with reduced protein yields upon extraction and affinity purification.

14.) The discussion section can be more concise in terms of polarity effects on detergent performance, since they are repeated in some parts without adding new insights.

We thank the referee for raising this point which helped us to improve the scientific value of our manuscript. We contextualized the discussion on detergent performance in protein purification and amplifying antimicrobial activities of antibiotics with literature to gain the following new insights: [...] Harsher detergents are more polar than milder detergents, which is reflected in reduced aggregation tendencies, higher cmc values, higher detergent monomer concentrations as well as higher overall detergent concentrations in assay buffers, which leads to more denaturing assay environments under the experimental conditions employed.^{21, 41}

Established chemical design strategies that can maximize cmc values include a reduction in the overall size of detergent molecules, a reduction in the length of the alkyl tail relative to a polar head, or a substitution of non-ionic triglycerol headgroups for ionic headgroups.³ Our data complement this knowledge by demonstrating that the utilities of equilibria between monomers and globular micelles formed by triglycerol detergents can be shifted from protein purification to mild amplification of antimicrobial activities of antibiotics by increasing cmc values through an increase in the polarity of the linker (Figure 5B). In line with our suggestion that non-ionic, worm-like-micelle-forming detergents do not efficiently interact with membrane components, we did not observe amplifications in antimicrobial activities of antibiotics with such detergents (Figure 5B). #

15.) Expand the potential reasons why harsh detergents perform better for antibiotic amplification but not for protein purification.

To expand the potential reasons why harsh detergents perform better for antibiotic amplification but not for protein purification, we contextualized our findings further with literature and obtained an additional knowledge gain, which we communicate as follows: "From a supramolecular perspective, most detergents provide antimicrobial properties in their monomeric form, e.g., below cmc,⁴⁹ because toxicity is linked to the insertion of monomeric detergents into membranes.⁵⁰ High cmc values lead to high monomer concentration around cmc whose uptake into membranes can facilitate structural perturbations in lipid bilayers and the diffusion of antibiotics into bacteria. This is the opposite to what is required for protein purification where detergents are used above cmc to encapsulate membrane proteins into detergent aggregates that secure protein solubility and stability.^{21, 41} Detergents used in the extraction and affinity purification of membrane proteins ideally have low cmc values to minimize protein denaturation.^{3, 21, 41} In line with this argumentation, our antibiotic amplification experiments on the model pathogen *E. coli* revealed that non-ionic detergents that are considered as "harsh" in protein purification, like detergents with shorter C8 alkyl chains and higher cmc values worked better in amplifying antimicrobial activities of antibiotics than detergents that are considered as "mild," like detergents with longer C12 alkyl chains and lower cmc values (Figure 3A-C). Harsher detergents are more polar than milder detergents, which is reflected in reduced aggregation tendencies, higher cmc values, higher detergent monomer concentrations as well as higher overall detergent concentrations in assay buffers, which leads to more denaturing assay environments under the experimental conditions employed.^{21, 41}"

16.) Lack of critical comparative analysis. While this manuscript cites previous studies, it lacks direct comparative data to emphasize novelty and contextualize findings.

We thank the referee for raising this critical point. Indeed, we cite previous studies and refer to protein purification data which we obtained by means of the same protein expression and purification protocol that we established before. To address this point, we revised the manuscript in the sub-section "Protein purification and supramolecular chemistry" as follows: "Interestingly, similar reductions in relative protein quantities were obtained for another model protein, i.e., GFP-tagged aquaporin Z (AqpZ-GFP), when the C12-alkyl chain in dendritic triglycerol detergents was displaced by (a) a C14-alkyl chain and (b) a partially fluorinated chain.⁷ Alternatively, a substitution of (c) a cholesterol tail in [G2] OGDs by a C12 double chain motif led to a similar observation (Supplementary Figure 2).⁵ All three modification, i.e., (a)-(c), led to detergents that formed worm-like micelles.^{5, 7, 38} Previously reported protein purification data on AqpZ-GFP^{5, 7, 38, 39} were done by following the same protocol as described here for MscL-GFP.³⁹ To exclude that our correlation is biased by the protein, herein, we repeated the purification of AqpZ-GFP under comparable conditions³⁹ with our detergents and obtained similar trends in relative protein quantities, which supports our conclusion that worm-like micelles correlate with low relative protein quantities (Supplementary Table 1) (Supplementary Figure 2 and 3) (Figure 3)."

In addition, the caption of Supplementary Figure 2 in the Supplementary Information was specified as follows to clarify where the reference data were taken from: "Bar charts visualize relative protein quantities of AqpZ-GFP obtained upon extraction and affinity purification with A) DTG detergents and B) second-generation DTG detergents. Detergents that form worm-like micelles gave lower protein yields, regardless of the structure of the head group and tail. Relative protein quantities shown in A) were taken from Ref.¹ and relative protein quantities shown in B) were taken from Ref.²" #

17.) Discuss potential limitations or next steps, such as scaling your methodology or testing against other Gram-negative pathogens

We address this point in the sub-section "Discussion" of the revised manuscript as follows: "Future studies need to clarify whether the incorporation of more polar linkers can also increase the cmc values and toxicities of other detergents classes. Further clarification is needed regarding the mode of action. Do detergents reduce the MIC of antibiotics more directly, for example, by promoting the diffusion of antibiotics into *E. coli*,⁵¹ or more indirectly, by destabilizing proton gradients across membranes and membrane protein machineries whose activities are coupled to proton gradients, such as observed before in the cases of certain amphiphilic efflux pump inhibitors.⁵² Future studies need to clarify if triglycerol detergents that efficiently solubilize lysed membrane fractions can synergize with reagents that solubilize outer membranes, such as benzalkonium chloride⁵³ or colistin.⁵⁴ Furthermore, since there is no such thing as a general cell wall structure, future studies need to contextualize the antibacterial properties of our detergents with different bacteria to better understand the impact of detergent design on selection pressure in microbial communities.¹⁹ A better understanding on how the structures of detergents affect their selectivity to different cell wall architectures could deliver new directions for the development of cell-selective amphiphiles that serve the purposes of applications but do not harm the wider biosphere in their roles as novel entities.⁵⁵

18.) In the conclusion section it would be interesting to summarize how the findings support future applications (e.g., drug delivery? or antibiotic formulations?).

To address this point, we revised the last concluding paragraph in the manuscript as follows: "Furthermore, our correlative study suggests that non-ionic detergents that are harsher in

protein purification work better in amplifying antimicrobial activities of antibiotics due to higher polarities, cmc values and detergent monomer concentrations under assay conditions. Taking together, these learnings represent a significant step forward for the design of detergents in supramolecular chemistry, membrane research, drug delivery, and antibiotic formulations. The modular chemistry of triglycerol detergents delivers a starting point for structure-activity relationships that will be widely deployed beyond membrane research.” #

Reviewer #3 (Remarks to the Author):

19.) Manuscript: Chemical linkers switch triglycerol detergents from bacterial protein purification to mild antibiotic amplification. The English language has to be improved through the manuscript. The authors discuss protein purification and mention membrane dissolution in the introduction section. However, they need to specify the type of protein they aim to explore—whether it is produced intracellularly by the microorganism or secreted into the culture medium.

We specified the type of protein and connection between protein purification and antimicrobial susceptibility assays as follows: “In the purification of inner or outer membrane proteins from *E. coli*, the solubilization of lysed membrane fragments with detergents is a key step to the extraction of high protein quantities into globular micelles.^{12, 20, 21} Whether detergent properties associated with satisfying protein yields following extraction and affinity purification from lysed *E. coli* membranes translate into potent antibiotic amplification on whole-cell *E. coli* remains to be addressed.”

20.) Additionally, different microorganisms have varying types of cell walls, which require distinct treatments and interactions with detergents for membrane disruption. I suggest revising the introduction section to clarify these points and provide more detailed explanations.

We thank the referee for raising this point, which we addressed in the discussion of possible future directions in our revised manuscript as follows: “Furthermore, since there is no such thing as a general cell wall structure, future studies need to contextualize the antibacterial properties of our detergents with different bacteria to better understand the impact of detergent design on selection pressure in microbial communities.¹⁹ A better understanding on how the structures of detergents affect their selectivity to different cell wall architectures could deliver new directions for the development of cell-selective amphiphiles that serve the purposes of applications but do not harm the wider biosphere in their roles as novel entities.⁵⁵”

21.) The size of the detergent tail may influence its interaction with the cell wall and the target molecule intended for purification. How does the authors' approach address or analyze this potential impact?

We aimed to address this impact by increasing the length of the alkyl chain in linear triglycerol detergents which led to detergents that were insufficiently soluble for our assays.

22.) The authors determined the critical micelle concentration (CMC) of the detergents, and I suggest including these results in a table for better clarity. Additionally, are there alternative methods to determine this parameter besides fluorescence? If so, it would be useful to discuss them.

To address this point, we show cmc values of detergents in Figure 3B along with relative protein quantities, aggregate morphologies, and observed antibiotic amplification factors. To better guide the readers to this information, we added a descriptive sentence to the sub-section “Protein purification and supramolecular chemistry” in the revised manuscript: “To investigate whether aggregate morphologies can be used to predict protein purification outcomes, we compared critical micelle concentration (cmc) values, aggregate morphologies obtained from our detergents above cmc (Supplementary Table 1) (Supplementary Figure 1), and relative

protein yields obtained upon extraction and affinity purification of the inner membrane protein mechanosensitive channel (MscL-GFP) (Figure 3A-C).”

Furthermore, we included a tabular summary with detergent names, cmc values, HLB values, diffusion coefficients, hydrodynamic diameters, and aggregate morphologies in the Supporting Information – see Supplementary Table 1.

We determined the cmc with as established protocol and refer to this in the revised Methods sections as follows: “The critical micelle concentration (cmc) was determined by means of an established solubilization assay including the model dye ‘Nile red’ and a fluorescence readout.⁵⁶” We feel expanding on the discussion of cmc techniques is beyond the scope of this work. #

23.) I recommend expanding the discussion section to highlight key factors that can guide the purification of a molecule using detergents. There are studies in the literature that report the extraction of molecules in micellar systems, yet the authors consistently use the term “purification” throughout the manuscript. Was this study designed to be applicable to other detergent families?

For details regarding key factors that guide the optimization of non-ionic detergents in biomolecule purification, please see answers to reviewer comments No. 14 and 15.

To clarify what is meant with purification, we rephrased the first sentence in our introduction as follows: “Saccharide detergents are gold standards for the extraction of membrane components into globular micelles and enable the affinity purification and structural analysis of membrane proteins by X-ray crystallography or cryo-electron microscopy.¹⁴”

Furthermore, the selection of assays utilized in this study is designed to be applicable to other detergent families. To clarify this point, we revised our conclusions as follows: “Taking together, these learnings represent a significant step forward for the design of detergents in supramolecular chemistry, membrane research, drug delivery, and antibiotic formulations. The modular chemistry of triglycerol detergents and complementary assay technologies deliver starting points for structure-property studies that will be widely deployed beyond membrane research.” #

24.) What challenges might arise when using different compounds? Furthermore, it would be important to clarify whether the detergents used in this study can remain associated with the molecule for its intended application, or if they must be removed. If removal is necessary, what procedures could be employed to achieve this effectively?

Regarding the question, whether it is important if the molecules remain associated, we added the following information to the sub-chapter “Discussion” in our revised manuscript: [...] This is the opposite to what is required for protein purification where detergents are used above cmc to encapsulate membrane proteins into detergent aggregates that secure protein solubility and stability while remaining bound to it.^{21, 41} Detergents used in the extraction and affinity purification of membrane proteins ideally have low cmc values to minimize protein denaturation.”